# Antimicrobial Drug-Resistant *Salmonella* in Urban Cats: Is There an Actual Risk to Public Health?

**DOI:** 10.3390/antibiotics10111404

**Published:** 2021-11-16

**Authors:** János Dégi, Kálmán Imre, Viorel Herman, Iulia Bucur, Isidora Radulov, Oana-Cătălina Petrec, Romeo Teodor Cristina

**Affiliations:** 1Faculty of Veterinary Medicine, Banat’s University of Agricultural Sciences and Veterinary Medicine Timișoara, Calea Aradului 119, 300645 Timișoara, Romania; viorelherman@usab-tm.ro (V.H.); iulia.bucur@usab-tm.ro (I.B.); oana.mateiu-petrec@usab-tm.ro (O.-C.P.); romeocristina@usab-tm.ro (R.T.C.); 2Faculty of Agriculture, Banat’s University of Agricultural Sciences and Veterinary Medicine Timișoara, Calea Aradului 119, 300645 Timișoara, Romania; isidora_radulov@usab-tm.ro

**Keywords:** *Salmonella*, zoonotic, cats, public health

## Abstract

The present study was undertaken to investigate the presence of *Salmonella* spp. in the faeces of client-owned cats in urban areas and to evaluate the risk that is posed to public health. Fresh faecal samples were collected directly from the rectums from 53 diarrhoeic and 32 non-diarrhoeic cats. The samples were individually screened for the presence of *Salmonella* spp. using standard methods and, in the case of positive findings, the resulting typical colonies were then biochemically confirmed using the VITEK^®^2 automated system. Subsequently, all of the *Salmonella* spp. isolates were molecularly tested for the presence of the *invA* gene. All of the isolates were serotyped using the slide agglutination technique according to the White–Kauffmann–Le Minor scheme. The phenotypic antimicrobial susceptibility profile of the isolated strains was obtained from the VITEK^®^2 system using specific cards from the Gram-negative bacteria. A total of 16 of the samples (18.82%) tested positive for *Salmonella* spp. according to conventional and molecular testing methods. Serotyping of the *Salmonella* isolates showed the presence of three serotypes, namely *S. enteritidis* (*n* = 9; 56.3%), *S. typhimurium* (*n* = 4; 25%), and *S. kentucky* (*n* = 3; 18.8%). All of the tested strains showed strong resistance towards cefazolin, cefepime, ceftazidime, and ceftriaxone. Additionally, resistance (listed in descending order of strength) was observed to trimethoprim/sulfamethoxazole (11/16; 68.8%), ampicillin (10/16; 62.5%), ampicillin/sulbactam (9/16; 56.3%), gentamicin (9/16; 56.3%), nitrofurantoin (8/16; 50.0%), and amikacin (5/16; 31.3%). No resistance was expressed against ciprofloxacin, ertapenem, imipenem, levofloxacin, piperacillin/tazobactam, and tobramycin. The results of this study highlight a substantial public health issue and medical concern, especially in vulnerable people, such as children, the elderly, and immunocompromised individuals.

## 1. Introduction

The need to treat cats that have infectious diarrhoea is a common problem for veterinarians. However, the role of enteropathogenic bacteria in this disease is poorly understood. It is not uncommon to find *Salmonella* spp. as culprits in feline diarrhoea, but their clinical evidence in cats is obscured by the fact that these bacteria are considered common to the indigenous intestinal microflora in many other animals [1]. Likewise, there are other zoonotic pathogenic agents in cats (e.g., *Clostridium perfringens* type A, *Clostridium difficile*, *Campylobacter upsaliensis*, *C. helveticus*, and *C. jejuni*) that may be responsible for conditions ranging from mild diarrhoea to fatal necroheamorrhagic enteritis [1,2,3,4]. Despite mounting concerns about other pathogens in recent years, *Salmonella* remains among the leading causes of food-borne disease worldwide [5,6]. 

Salmonellosis involves a broad spectrum of diseases in humans and animals; it is able to manifest as acute gastrointestinal enteritis, bacteriemia, and extra-intestinally localised infections involving many organs [7]. In humans, infections caused by *Salmonella* spp. are associated with severe food-borne illnesses, especially in the case of acute gastroenteritis, which is caused by consuming contaminated water and food products [8,9].

Cats and dogs are the most widely kept pet animals. However, the *Salmonella* carriage status of these animals is largely unknown, and the posed risk to the owner’s health is unclear [10,11]. In particular, cats that can freely roam outside and can scavenge or hunt for food of unknown quality are potential candidates for *Salmonella* spp. carriage [10]. Subclinical infections in carrier animals can lead to the transmission of the bacterium to humans, which is a much more critical concern [12,13].

The increased popularity of high-protein raw meat-based diets (RMBDs) demonstrates the growing preference for higher protein quantity and quotient in pets’ diets. Uncooked animal products, such as skeletal muscle, fat, internal organs, cartilage, bones (from ruminants, pigs, poultry, horses, game, or fish), unpasteurised milk, and uncooked eggs are included in these diets [14]. The macronutrient profiles of these diets differ from those of commercial pet foods, with RMBDs containing higher protein and fat content. As a result of the potential for pathogen contamination, RMBD diets are not generally recommended by most major veterinary and public health organisations [15]. However, they are becoming increasingly popular. The GI microbiomes and metabolisms of animals fed RMBDs are different from those fed extruded, heat-processed foods, and this has been shown in numerous studies [16,17,18,19]. Feeding raw foods may increase the risk of exposure to potentially pathogenic bacteria for companion animals and human subjects. During infection, pathogenic bacteria compete with and displace the commensal species, resulting in the dysbiosis of the microbial population and gastrointestinal upset [17].

Year by year, the increased multidrug resistance of zoonotic pathogens, including *Salmonella,* is recognised as one of the most important public health concerns in all geographic regions. This phenomenon is strongly related to the misused and prolonged antibiotic treatments prescribed in human and veterinary medicine [20,21,22].

During the last decade, and under the guidance of the One Health approach, significant signs of progress have been made in the monitoring of the occurrence rate and antimicrobial susceptibility level of *Salmonella* isolates that have been detected within the food chain [23,24,25,26]. However, no information is available concerning cats’ *Salmonella* carriage status, and that of other pets and livestock in general, in Romania. Thus, this study aimed to provide data on *Salmonella* strains’ occurrence and their antimicrobial susceptibility profiles in client-owned cats living in urban areas, and to evaluate the risk posed to public health in Timișoara, one of the major urban settlements in western Romania.

Overall, the collaboration between veterinary and medical professionals and widespread, cooperative efforts to enhance health education are helping to reduce the risks of food-borne and zoonotic infections and supporting the achievement of the goals outlined in the One Health approach [27]. In addition to these concerted efforts, the risk factors require frequent re-evaluation due to the close human–animal bonds shared with pet animals, the changed human–companion animal bond, the many evolving recommendations regarding responsible pet ownership (including standard hygiene practices, responsible breeding, feeding, housing requirements, and mental stimulation), and the physical challenges concerning the biology of animals [28].

## 2. Results

Overall, a total of 16 out of the 85 investigated cat faecal samples (18.82%) tested positive for *Salmonella* spp. Using conventional techniques, all of the identified *Salmonella* strains were successfully amplified within PCR reactions that targeted the *invA* gene (~284 bp). In the present study, the highest transmission rate of infection was recorded in the 3 to 6 years of age category (8.23%). The *Salmonella* carriage status of cats, organised according to their sex and age group, is summarily presented in Table 1. No statistically significant associations (*p* < 0.05) were found between the infection status and any of these epidemiological factors. In agreement with the obtained results, there have been no reports in similar studies on the influence of gender on the prevalence of *Salmonella* strains isolated from cats. However, other studies have shown that younger and older animals are more likely to acquire pathogens after exposure, including *Salmonella*, which can result in a severe infection [29].

Table 2 presents the results of species identification using the ID-GNB card. Accordingly, the identified serotypes were categorised into several confidence levels: excellent, very good, acceptable, good, low, unidentified, and error. In this regard, three serovars were identified: S. *enteritidis* (9/16; 56.3%), *S*. *typhimurium* (4/16; 25%), and *S. kentucky* (3/16; 18.8%).

Previously conducted studies highlight that several *Salmonella* serovars have been identified in cats, the appearances of which have been found to vary according to their geographic region. Generally, these studies’ results have noted that the serovars which are more commonly found in cat faeces tend to be the same as those found in humans [30,31]. In this study, a total of three serovars were identified, and they correspond to those that have been most frequently reported by the Centers for Disease Control and Prevention (CDC) in humans within European countries [31]. Salmonellosis is a major zoonosis, for which there have been numerous historical reports that indicate the transmission of the pathogen through food, farm animals, and from pets [12,13]. In this regard, multiple cases have been published describing infections with the same transmission route, between humans and animals, at the household level [13]. The transmission route remained unclear in many cases, but the transmission modalities from food to animals or humans, or between humans and animals, have been frequently documented. Therefore, it is essential to evaluate the risk factors that increase the likelihood of an infection.

The evidence of somatic (O) and flagellar (H) antigen results from the slide agglutination technique that was performed on the isolated Salmonella strains is summarised in Table 3. 

In the present investigation, a positive association (*p* < 0.005) was found between the diets of cats consuming raw food, especially raw meat, and the presence of *Salmonella* spp. in their faecal samples (Table 4). Study results from tests on cat food for the presence of *Salmonella* spp. have underlined that, over time, the occurrence of *Salmonella* spp. in marketed dry diets has decreased considerably, most likely due to better controls within the manufacturing processes [32]. 

The survey results demonstrated that *Salmonella*-positive faecal samples from the outdoor cats accounted for 50% of all positive results (8/16). According to the results published in the scientific literature, outdoor cats are more likely to test positive for *Salmonella* spp. bacteria [33]. In a study conducted by Thomas et al. [34], the diversity and seasonality of the *Salmonella* serotype in urban and rural cats were investigated. The study results showed that urban cats had a higher prevalence of salmonellosis than rural cats.

Other factors that can increase the spread of *Salmonella* infections include improper food handling and faster growth of the bacteria during the summer season. These were considered the primary risk factors in the aforementioned study. Similarly, the change in owners’ culinary preferences resulting in increased consumption of grilled meat during summer can subsequently increase the consumption of uncooked meat, favouring the spread of the infection.

All 16 of the isolated *Salmonella* strains were found to be resistant to at least three antimicrobial agents. The isolated strains showed many resistances; in particular, many were resistant towards cefazolin, cefepime, ceftazidime, and ceftriaxone (Table 5). In addition, resistance was observed (listed in descending order of strength) towards other antimicrobials, including trimethoprim/sulfamethoxazole (11/16), ampicillin (10/16), ampicillin/sulbactam (9/16), gentamicin (9/16) nitrofurantoin (8/16), and amikacin (5/16). There was no resistance found to the following antimicrobials: ciprofloxacin, ertapenem, imipenem, levofloxacin, piperacillin/tazobactam and tobramycin. 

The year-by-year increase in the antimicrobial resistance phenomenon in the zoonotic bacteria, including *Salmonella*, within the human–animal food chain is considered a severe public health concern. However, the susceptibility of *Salmonella* to antimicrobial drugs can vastly vary among countries and between regions [20,21,22].

For veterinarians and cat owners alike, the diagnosis of diarrhoea is frustrating as it can be caused by extra-intestinal diseases or gastrointestinal diseases (for example, dietary causes, gastrointestinal infection, inflammation, or neoplasia) [35]. Feline salmonellosis is traditionally diagnosed using the isolation of a *Salmonella* spp. strain in conjunction with observation of the relevant clinical signs and an assessment of the potential risk factors. This is because the presence of *Salmonella* bacteria in the cat may not be sufficient for a salmonellosis diagnosis when observed on its own [36]. Salmonellosis is suspected in our study based on the culture results, the clinical symptoms, and the discovery of *Salmonella* spp. DNA in the cats’ faeces. 

## 3. Discussion 

Our study agrees with the conclusions of many previous investigations [29,37,38]. Some factors must also be considered when analysing the prevalence of *Salmonella* isolates in different regions; these factors include pet hospital management, the reproductive status of the animal, the sampling method used, the season in which the sampling took place, and the method used for the isolation of the bacteria [39]. According to Wei et al. [33], cats had a prevalence of 1.77 per cent for *Salmonella* spp. isolates. *Salmonella kentucky* (*n* = 11), *Salmonella enterica subsp. enterica* (*n* = 8), *Salmonella indiana* (*n* = 5), and *Salmonella typhimurium* (*n* = 4) were the most frequent serotypes to be isolated from cats’ samples. According to a study published by Reimschuessel et al. [29], the prevalence of *Salmonella*-positive dogs and cats has declined in recent decades. Our study also indicates that raw food consumption is a significant risk factor for *Salmonella* infection. It is worth noting that nearly half of the animals that tested positive for *Salmonella* were otherwise asymptomatic (i.e., they were non-diarrhoeic).

The occurrence rate of *Salmonella* isolates in cats has been established in many other studies that were carried out in numerous geographic regions. The most prevalent species found in this study, adjacent *Salmonella enteritidis* and *Salmonella typhimurium*, have also been reported in previous studies presented by Van Immerseel et al. [10] and Guardabassi et al., [40] and, more recently, by Reimschuessel et al. [29] and Wei et al. [33]. 

Salmonellosis is uncommon in cats; however, no data or research estimating feline salmonellosis or large-scale epidemiological studies examining *Salmonella* risk factors are available. [41]. Wild birds were implicated in a *Salmonella typhimurium* infection outbreak in cats and humans, during which salmonellosis was thought to have been transferred from cats to humans [42].

In the case of the *S. typhimurium* strain, pets may serve as foci of the nosocomial transmission of *Salmonella* between animals and humans if the appropriate precautions are not followed [43]. Salmonellosis is most commonly found latently in dogs and cats [44]. Different pathogen counts, the host immune status, health complication occurrences, and different disease unit forms lead to different clinical findings. 

In the present study, antibiotics that are used to treat infections in humans and pets were found to face high levels of resistance from some isolated strains of *Salmonella*; this raises the possibility that humans could become infected with multidrug-resistant *Salmonella* through contact with cats. As a valuable class of antibiotics for treating various human and animal infections, fluoroquinolones are particularly effective against salmonellosis. To keep fluoroquinolones as effective as possible, it is important to use them carefully, regularly check for antibiotic residues in food, and provide comprehensive monitoring for the emergence of bacterial resistance in both animals and humans [45]. Carbapenems (ertapenem and imipenem), known as the “last resort” antibiotics for use in cases that have required drug resistance monitoring, are required to establish any possible links between reservoirs of bacteria and to limit the bidirectional transfer of the encoding genes between *Salmonella* spp. and other commensal or pathogenic bacteria. *S. enterica* has a clinical impact as a nosocomial pathogen in humans and the frequency with which it is found to have a resistance to both of the carbapenems is modest compared to other *Enterobacteriaceae* [46]. Nonetheless, there have been numerous serovars derived from human clinical samples, including S. *typhimurium* and *S. kentucky*, that have been found to produce carbapenemase. *S. typhimurium* and *S. kentucky*, two other pathogens with high levels of multidrug resistance (MDR), are linked to the spread of virulent clones [46]. Human infections in the European Union have been linked to these three serovars more often than any other geographic group [47]. 

The diarrhoea caused by *Salmonella* spp. in cats is frequently difficult to distinguish from diarrhoea caused by other bacteria. The disease manifests itself in mild to severe gastroenteritis, depending on the individual. It is challenging to diagnose bacteria-associated diarrhoea in cats because there is no objective guidance for faecal testing, and identical isolation ratios for the presumed bacterial enteropathogens have been discovered in populations of animals both with and without diarrhoea [36].

According to results of several of the studies from the scientific literature [36,48], the prevalence of *Salmonella* spp. in cats varies and is similar in diarrhoeic and non-diarrhoeic cats, with shedding rates ranging from 0 to 8.6% in diarrhoeic cats and from 0 to 14% in non-diarrhoeic cats.

Raw food diets for cats are becoming increasingly popular, and there are various options available to cat owners who want to change their feeding practices. When eating a raw diet, there is a genuine risk of contracting well-known bacteria such as *Salmonella* [41]. The most common sources of *Salmonella* spp. for indoor cats are considered to be the raw meat provided to them and some processed foods, whereas outdoor cats are at risk of infection from scavenging and hunting rodents and birds, exposure to reptiles, and environmental contamination [41,49].

The dangers of raw feeding are real, especially if pet owners cannot prepare complete, balanced, and safe meals for their animals, or if they do not purchase and store raw meals properly. Raw food is well known to pose a significant risk of infectious disease to the pet, the pet’s environment, and the pet’s owner [50]. Further studies investigating the microbiological safety of raw foods used in the feeding of cats in the investigated region are still necessary for an objective evaluation of the risk that is posed.

## 4. Conclusions

The results of the present study demonstrate that the client-owned cats that were included in this analysis may be considered an important *Salmonella* reservoir in the investigated region, potentially excreting the pathogen through their faeces and contaminating the environment. Additionally, both cats with diarrhoea and clinically healthy ones could threaten public health, especially the health of children and immunocompromised people. Following good hygiene practices and minimising the transmission risk of zoonotic infections are recommended. Furthermore, the worrying multidrug resistance of the isolated strains identified in this study highlights the urgent need for the implementation of efficient antimicrobial stewardship programs, due to the ongoing need to protect both human and animal health.

## 5. Materials and Methods

### 5.1. Sample Collection

Between April and September 2019, a total of 85 faecal samples derived from client-owned cats originating from Timișoara Municipality, western Romania, were collected and screened in order to estimate the prevalence of drug-resistant *Salmonella* spp. The enrolled animals had presented for veterinary services at the University Veterinary Clinics of the Faculty of Veterinary Medicine, Timișoara. 

All sampling protocols were followed in accordance with the relevant national guidelines and regulations. The collection of the faecal specimens was performed with the consent of the pets’ owners, according to the code of the Romanian Veterinary College (protocol numbers 34/1.12.2012) and the procedures of the University Veterinary Clinics of the Faculty of Veterinary Medicine Timisoara. 

A common procedure was established to recruit the diarrhoeic and the non-diarrhoeic cats to the study. The diarrhoeic cats (*n* = 53) were presented by the owner to the veterinarian with a current gastrointestinal disorder. The faecal consistency was determined, based on a faecal scoring system where a score of 1 was considered very firm, a score of 2 was considered well-formed, a score of 3 was considered soft formed, and a score of 4 was considered watery [51]. Fresh faecal samples from apparently clinically healthy, non-diarrhoeic cats (*n* = 32) were obtained from animals belonging to veterinary students, during different deworming or immunisation actions. The faecal samples were collected in sterile plastic containers directly from the rectum or as a consequence of spontaneous emission. The harvested specimens were stored in refrigerated boxes and transported to the laboratory in the shortest time possible. During sampling, the owners provided information about the cats’ ages (≤3 yr, from 3 to 6 yr, or older than 6 yr), gender (male or female), breed (Persian, Russian Blue, Siamese, British Shorthair, American Shorthair, Burmese, Maine Coon, or European mix), lifestyle (indoor, outdoor, or partial outdoor), and type of diet (commercial or raw meat diet). Detailed information about these data is presented in Table 1 and Table 4.

### 5.2. Bacterial Isolation

*Salmonella* spp. bacteria were isolated using conventional methods and following the protocols recommended by Food and Drug Administration (FDA), the Commission Regulation (EC) No 2073/2005 [52], and according to the International Standard Organization (ISO) 6579:2002 standard, revised by ISO 6579-1:2017 [53]. The samples were processed on the day of sampling in the Bacterial Diseases Diagnostic Laboratory (B.6.a), of the Department of Infectious Diseases and Preventive Medicine, located in the Faculty of Veterinary Medicine, Timișoara.

Firstly, 25 g of each collected sample was homogenised with 10 mL of selenite cystine broth (BBL Selenite-F Broth, Thermo Fisher Scientific, Waltham, MA, USA) for 5 min. The mixture was then incubated at 35 °C in an aerobic atmosphere for 24 h. Next, one millilitre from each of the pre-enriched samples, after the incubation period, was added to 10 mL of Rappaport Vassiliadis broth (Thermo Fisher Scientific, Waltham, MA, USA), and mixed for 2 min, then subsequently incubated in an aerobic atmosphere for 24 h at 42 °C. After incubation, the tube content was stirred and inoculated into the MacConkey agar plates (Thermo Fisher Scientific, Waltham, MA, USA) with a bacteriological inoculation loop. The inoculated MacConkey agar plates were incubated at 35 °C in an aerobic atmosphere for 24 h. After this stage, presumptive *Salmonella* spp. colonies with a characteristic morphology on the XLD agar (Thermo Fisher Scientific, Waltham, MA, USA) media were tested for their biochemical characteristics, including xylose fermentation, lysine decarboxylation, and production of hydrogen sulphide. The plates were further incubated at 37 °C in an aerobic atmosphere for 24 h.

*Salmonella* species were identified with the VITEK^®^2 automated compact system (bioMérieux, Marcy l’Etoile, France) using the AST–Gram-negative specific bacteria card, which is designed for the automated identification of most clinically significant fermenting and nonfermenting Gram-negative bacilli [54].

### 5.3. Molecular Analyses

All biochemically identified *Salmonella* strains were directly subjected to molecular analysis. Bacterial genomic DNA was isolated from the strains cultivated on selenite cystine broth media (BBL Selenite-F Broth, Thermo Fisher Scientific, Waltham, MA, USA) using a PureLink™ Genomic DNA Mini Kit (Thermo Fisher Scientific, Waltham, MA, USA), according to the manufacturer’s specifications. The extracted DNA quantity and quality was determined using a NanoDrop ND-1000 spectrophotometer (NanoDrop^®^ Technologies, Thermo Fisher Scientific, Waltham, MA, USA) by measuring the absorbance at 260 nm. The strains were then molecularly tested for the presence of the *Salmonella*-specific *invA* gene (~284 bp) by using a conventional polymerase chain reaction, as was previously described by Lampel et al. (2000). Specific forward (5′ GTG AAA TTA TCG CCA CGT TCG GGC AA-3′) and reverse (5′ TCA TCG CAC CGT CAAAGG AAC C-3′) primers were used [55]. The PCR conditions consisted of an initial denaturation at 95 °C for 5 min, followed by 32 cycles of denaturation at 95 °C for 1 min, annealing at 55 °C for 1 min, extension at 72 °C for 1 min, and a final extension at 72 °C for 10 min, using the *My Cycler* (BioRad^®^, Dubai, United Arab Emirates) thermocycler. All PCR amplicons were visualised on ethidium bromide-stained 2.5% agarose gel under UV light using a gel documentation system (UV transilluminator–2035-2, Bio Olympics USA). The strain *Salmonella enterica* serovar, ATCC 13076, was used as positive control. The negative control consisted of sterile deionised water.

### 5.4. Serotyping by Slide Agglutination (Kauffmann–White–Le Minor Scheme)

Serotyping of the *Salmonella* isolates was achieved in a pure culture based on the evidence of somatic (O) and flagellar (H) antigens through reactions with specific antisera [56]. In this regard, the BD Difco Salmonella O and BD Difco Salmonella H antisera (Becton Dickinson and Company, Franklin Lakes, NJ, USA) were used, in accordance with the manufacturer’s recommendations.

### 5.5. Antimicrobial Susceptibility Testing

The antimicrobial susceptibility testing of the isolated *Salmonella* strains was performed using the VITEK^®^2 testing system and the AST GN67 card (bioMérieux. Marcy l’Etoile, France). 

The tested antimicrobials were: amikacin (AN; MIC range 16–64 μg/mL), ampicillin (AM; MIC range 8–32 μg/mL), ampicillin/sulbactam (SAM; MIC range 8/4–32/16 μg/mL), cefazolin (CZ; MIC range 2–8 μg/mL), cefepime (FEP; MIC range 2–16 μg/mL), ceftazidime (CAZ; MIC range 4–16 μg/mL), ceftriaxone (CRO; MIC range 1–4 μg/mL), ciprofloxacin (CIP; MIC range 0.06–1 μg/mL), ertapenem (ETP; MIC range 0.5–2 μg/mL), gentamicin (GM; MIC range 4–16 μg/mL), imipenem (IPM; MIC range 1–4 μg/mL), levofloxacin (LEV; MIC range 0.12–2 μg/mL), nitrofurantoin (FT; MIC range 32–128 μg/mL), piperacillin/tazobactam (TZP; MIC range 16/4–128/4 μg/mL), tobramycin (TM; MIC range 4–16 μg/mL), and trimethoprim/sulfamethoxazole (SXT; MIC range 2/32–4/76 μg/mL). The obtained results were automatically processed by the system, and the isolates were categorised as susceptible, resistant, or intermediate. The isolates resistant to three or more classes of antimicrobials were classified as multidrug resistant.

### 5.6. Statistical Analysis

The obtained results were statistically interpreted using the SPSS statistical analysis software package, version 21.0. A nonparametric Pearson’s chi-squared (χ^2^) test was used in order to find any possible associations between the *Salmonella* infection status and the recorded epidemiological data. Differences were established as statistically significant when *p*-value  ≤  0.05.

## Figures and Tables

**Table 1 antibiotics-10-01404-t001:** Distribution of the isolated *Salmonella* strains according to age and gender.

Parameters	No. of Faecal Samples Collected	Positive Coproculture from *Salmonella* spp.	
	*n*	*n*	%
Age (years)			
≤3	28	5	5.88
from 3 to 6	53	7	8.23
>6	25	4	4.71
Total	85	16	18.82
Gender			
Female	47	6	7.05
Male	38	10	11.77
Total	85	16	18.82

**Table 2 antibiotics-10-01404-t002:** Results showing *Salmonella* spp. identification and their probability level from the VITEK^®^2 ID-GNB identification card (bioMérieux. Marcy l’Etoile, France).

Identified Strains	No. (%) of Strains Identified at the Following Probability Level							
	N (%)	Excellent	Very Good	Acceptable	Good	Low	Unidentified	Error
*Salmonella enteritidis*	9 (56.25%)	7	2	-	-	-	-	-
*Salmonella typhimurium*	4 (25.00%)	1	3					
*Salmonella kentucky*	3 (18.75%)	2	1					
Total	16	10	6	-	-	-	-	-

**Table 3 antibiotics-10-01404-t003:** Somatic and flagellar antigen distribution within the 16 isolated *Salmonella* strains.

Serotypes	O-Antigens	H-Antigens	Number of Isolates
*Salmonella enteritidis*	1, 9, 12	f, g, m, p 1,7	9
*Salmonella typhimurium*	1, 4, 5, 12	I, 1, 2	4
*Salmonella kentucky*	8, 20	I, z6	3

**Table 4 antibiotics-10-01404-t004:** Distribution of cat *Salmonella* spp. positive faecal samples, according to diet, habitat, and clinical aspects.

Laboratory Result	Nutrition			Habitat			Clinical Signs	
	Commercial Food		Raw Meat Diet	Outdoor	Indoor	Partial Outdoor	Diarrhoeic	Non-Diarrhoeic
	Dry	Wet						
	21	18	46	25	54	6	53	32
Total samples	85			85			85	
Positive *Salmonella* spp. samples	1 (6.25%)	5 (31.25%)	10 (62.50%)	8 (50.00%)	6 (37.50%)	2 (12.5%)	9 (56.25%)	7 (43.75%)
Total	16			16			16	

**Table 5 antibiotics-10-01404-t005:** Antimicrobial drug resistance profile of the isolated cat-origin *Salmonella* strains.

No.	Antibiotics	*Salmonella* Serotype		
		*Salmonella typhimurium* (*n* = 4)	*Salmonella enteritidis* (*n* = 9)	*Salmonella kentucky* (*n* = 3)
1.	amikacin (AN)	0	4	1
2.	ampicillin (AM)	0	8	2
3.	ampicillin/sulbactam (SAM)	0	8	1
4.	cefazolin (CZ)	4	9	3
5.	cefepime (FEP)	4	9	3
6.	ceftazidime (CAZ)	4	9	3
7.	ceftriaxone (CRO)	4	9	3
8.	gentamicin (GM)	4	4	1
9.	nitrofurantoin (FT)	0	5	3
10.	trimethoprim/sulfamethoxazole (SXT)	3	7	1

## Data Availability

The datasets generated and analysed during the current study are included within the article.

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
