# Peer review of "Antimicrobial Drug-Resistant Salmonella in Urban Cats: Is There an Actual Risk to Public Health?"

_antibiotics, 2021, doi:10.3390/antibiotics10111404_

Round 1
Reviewer 1 Report
I am very glad to review this manuscript and I would like to thank the editors for the indication.
I believe the work has great importance in general public health. Because of this, I suggest some improvements.
In general terms, the writing and grammar are very good, but there are some few mistakes that need to be double checked. For instance, in page 4, there are two sequenced paragraphs that begin with "In our study" and I think there is a "as" missing in the ending of line 130, in "" ... spreading of Salmonella infections, AS improper food handling...". So, I suggest a review of the english of the manuscript.
Unless the presentation of tables is considered format-free by the journal, ideally, the tables are usually presented as ilustrated in this link https://projects.ncsu.edu/labwrite/graphics/Example4.gif. Note that there are only the horizontal lines in the academic conception of tables.
Regarding to major questions:
1. Why the method section are in the ending of the manuscript? Unless this is a requirement of the journal, I strongly suggest the methodology immediately before the introduction section. In this kind of presentation, the results is not well conected with the remain text and there are lose of sense. In addition, it is not practical for the reader.
2. In page 2, line 74, I suppose you mean "peak of infection" as the group in which there is the highest prevalence. However, I believe there may be other interpretation for this expression, for example, the moment when the infected individual presents the maximum level of clinical signs, or when he/she presents the highest transmission rate. Thus, I suggest you use other expression or give previously to the reader your interpretation for "peak of infection".
3. The table 5 is one of the most important in your manuscript, but in my opinion it is not properly presented. I suggest you use a contingency table, where you consider 11 rows (each row is a antimicrobial) and 3 collumns (each collumn is a Salmonella serotype). In this kind of table, it is possible to clearly indicate the quantity of positive samples that presented resistace for each antimicrobial. You can find an example of contingency table here https://i.stack.imgur.com/L5MhX.png
4. Maybe I am wrong, but I believe the discution stressed the interpretation about the association among the positivity/negativity of the individuals and the other variables (as, nutrition, habitat, clinical signs, so on). However, I understand that the epidmiology of this disease (Salmonellosis in cats) may be much more complex than this. Thus, I suggest you explore more the possibilities of association among the variables (that is, a multivariate approach). For example, what are your perceptions about the clinical signs, habitat and positivity/negativity? Are there any epidemiological detail that could be considered as a novel in the scientific literature? I believe the variables of your dataset can be much more explored.
Author Response
I want to thank them for their valuable comments and suggestions, which have been of help to improve the quality of the manuscript. In this respect, we provide a Point-by-Point response to all posed questions and remarks as the R1- Final revision. To be easy findable we marked all our answers/corrections in yellow.

Reviewer 2 Report
I wonder why the study is limited to Salmonella, authors have anyway collected fecal samples, and it would have been nice to check the whole microbiome. The data would have given a broad understanding of the cat microbiome of the particular region.
Overall, the cat foods or raw foods fed to cats would be more or less similar in a given area. So, understanding microbiome shifts could have added more to the manuscript. The manuscript is well written; nevertheless, grammatical errors need to be minimized.
Introduction:
There are other multi-drug zoonotic pathogens apart from Salmonella. So, authors should give the background of different strains and then highlight the importance of Salmonella.
Results and Discussion:
I would appreciate it if the authors could separate the discussion from the results and elaborate on the discussion part.
In addition, add a phylogenetic tree of isolates.
Avoid the use of personal pronouns. There are too many “Our” in the text.
Line 221: can > may be
Line 213: can > could
Author Response
I want to thank them for their valuable comments and suggestions, which have been of help to improve the quality of the manuscript. In this respect, we provide a Point-by-Point response to all posed questions and remarks as the R1- Final revision. To be easy findable we marked all our answers / corrections in yellow.

Round 2
Reviewer 2 Report
The authors addressed my concerns, and have revised the manuscript as requested. However, I would suggest merging small paragraphs.
Best wishes,